# Investigation of Antioxidant Activity of Protein Hydrolysates from New Zealand Commercial Low-Grade Fish Roes

**DOI:** 10.3390/md22080364

**Published:** 2024-08-11

**Authors:** Shuxian Li, Alan Carne, Alaa El-Din Ahmed Bekhit

**Affiliations:** 1Department of Food Sciences, University of Otago, P.O. Box 56, Dunedin 9054, New Zealand; lish6041@student.otago.ac.nz; 2Department of Biochemistry, University of Otago, P.O. Box 56, Dunedin 9054, New Zealand; alan.carne@otago.ac.nz

**Keywords:** antioxidant activity, hydrolysate, protease, Hoki roe, gemfish roe, value-adding, delipidation, freeze-drying, nutrient composition

## Abstract

The objective of this study was to investigate the nutrient composition of low-grade New Zealand commercial fish (Gemfish and Hoki) roe and to investigate the effects of delipidation and freeze-drying processes on roe hydrolysis and antioxidant activities of their protein hydrolysates. Enzymatic hydrolysis of the Hoki and Gemfish roe homogenates was carried out using three commercial proteases: Alcalase, bacterial protease HT, and fungal protease FP-II. The protein and lipid contents of Gemfish and Hoki roes were 23.8% and 7.6%; and 17.9% and 10.1%, respectively. The lipid fraction consisted mainly of monounsaturated fatty acid (MUFA) in both Gemfish roe (41.5%) and Hoki roe (40.2%), and docosahexaenoic (DHA) was the dominant polyunsaturated fatty acid (PUFA) in Gemfish roe (21.4%) and Hoki roe (18.6%). Phosphatidylcholine was the main phospholipid in Gemfish roe (34.6%) and Hoki roe (28.7%). Alcalase achieved the most extensive hydrolysis, and its hydrolysate displayed the highest 2,2-dipheny1-1-picrylhydrazyl (DPPH)˙ and 2,2′-azino-bis(3-ethylbenzothiazoline-6-sulfonic acid) (ABTS) radical scavenging activities and ferric reducing antioxidant power (FRAP). The combination of defatting and freeze-drying treatments reduced DPPH˙ scavenging activity (by 38%), ABTS˙ scavenging activity (by 40%) and ferric (Fe^3+^) reducing power by18% (*p* < 0.05). These findings indicate that pre-processing treatments of delipidation and freeze-drying could negatively impact the effectiveness of enzymatic hydrolysis in extracting valuable compounds from low grade roe.

## 1. Introduction

Edible fish roes have been considered a high-value nutritious food due to their high content of high-quality protein, unsaturated fatty acids, vitamins, trace elements, and unique sensory attributes [1]. Physical damage due to rough handling causes a substantial portion of harvested roes fail to reach a desirable quality grade and is not profitable to process. Enzymatic hydrolysis to generate bioactive peptides is a potential method to increase the value of low-grade fish roe since the intactness of the roe is not technical or quality issues. Several studies reported that protein hydrolysates derived from fish roe exhibit notable bioactivities, such as ACE inhibitory peptides discovered in *Salmo salar*, *Carassius gibelio*, and collagens of Monkfish (*Lophius litulon*) Swim Bladders as well as antioxidant peptides found in the Skipjack tuna (*Katsuwonus pelamis*) roe [2,3,4].

In vitro enzymatic hydrolysis has been extensively used to generate bioactive peptides from high-protein foods and enhance their functional properties while preserving their nutritional values. Specifically, hydrolysis using proteolytic enzymes such as Alcalase, pepsin, trypsin, neutrase, and papain has shown a promise in producing bioactive protein hydrolysates from fish roe [5,6]. Common carp (*Cyprinus carpio*) roe is often treated as waste or animal feed due to its low-value and sensory appeal [7]. Therefore, it has been widely studied for the production of roe hydrolysate, and these studies reported good emulsion activity, forming capacity, and antioxidant activity of the obtained hydrolysates [7,8]. Additionally, enzymatic hydrolysis of roe of other commercial fish species has revealed important bioactivities such as immunomodulatory activity and peptidase IV inhibition, potentially regulating insulin secretion, and controlling hyperglycemia [9,10], which supports a potential use of low-grade roes.

Enzymatic hydrolysis can be performed using either a single enzyme or a combination of multiple enzymes as a pre-treatment to liberate bioactive compounds more efficiently from natural by-products [11]. Many studies have shown that using proteases such as Alcalase, pepsin, and trypsin to hydrolyze delipidated and freeze-dried fish roe can produce peptides with antioxidant activity, antibacterial properties, anticancer properties and enzyme inhibitory effects [2,12]. In the present study, a new commercial bacterial protease, HT, and a fungal protease, FP-II were investigated for their capabilities to produce antioxidant fish roe hydrolysates from Gemfish and Hoki roes and Alcalase was used alongside as a reference enzyme for comparative purposes. The food-grade protease preparations HT and FP-II have previously demonstrated effectiveness in hydrolyzing proteins from Hoki roe, beef myofibrillar, and connective tissues [13,14], as well as in producing bioactive peptides from by-products of the meat industry [15]. However, the impacts of delipidation and freeze-drying treatments on the hydrolysis and antioxidant activity of these specific protease preparations are yet to be assessed.

## 2. Results and Discussion

### 2.1. Proximate Composition

The proximate compositions (moisture, crude protein, crude fat, ash, and carbohydrate contents) of grade 4 Hoki and Gemfish roes are shown in Appendix A. There was no difference in ash content between Hoki and Gemfish roes (*p* > 0.05). Hoki roe had a higher moisture content (68.8%) and lipid content (10.1%) than Gemfish roe (*p* < 0.05). Higher protein content (23.8%), and carbohydrate content (4.2%) were found in Gemfish roe (*p* < 0.05). Bekhit et al. [16] reported that the moisture, crude protein, crude fat, ash, and carbohydrate contents of grade 1 Hoki roe are 64.7%, 20.2%, 10.9%, 2.1%, and 2.2%, respectively. Bah et al. [17] also found that grade 1 Hoki roe has 62.6% moisture, 20.6% protein, 11.0% lipid, 4.2% carbohydrate, and 1.6% ash content. Compared to grade 1 Hoki roe, grade 4 Hoki roe has similar lipid and ash contents and lower protein contents. The majority of fish roe products such Cod, Capelin, Lumpfish, Skipjack, Tongol and Bonito roes have lower values of about 3.4–5.7% and few such as Salmon, trout and Pike (8.4–10.6%) have relatively higher total crude lipid content [4,17,18,19]. The moisture and the protein contents of Hoki and Gemfish roes are within the moisture and protein ranges of raw roes (45.1% to 71.3% and 17.92 to 23.78%, respectively) reported by Bunga et al. [1]. Fish roe generally contain relatively low ash and carbohydrates, but Gemfish roe has relatively high carbohydrate content compared to other roes reported to have less than 3.0% carbohydrate content [16].

### 2.2. Fatty Acid Composition

The fatty acid compositions of Hoki and Gemfish roes are shown in Table 1. Most of the fatty acid contents found in the Hoki and Gemfish roes were different. C14:1 and C18:2 tn-6 were not detected in the Gemfish roe. Monosaturated fatty acids (MUFA) were the most abundant lipid group in both roe samples (1881.3–2783.7 mg/100 g of fresh weight, and 40.2–41.5% of total fatty acid). The polyunsaturated fatty acids (PUFA) content was in the range of 1591.9–2180.2 mg/100 g of fresh weight, and 31.8–35.8% of total fatty acid. Based on 100 g of fresh roe material, Hoki roe has higher MUFA, PUFA and saturated fatty acid (SFA) content (1631.3 mg/100 g of fresh weight) than Gemfish roe (947.2 mg/100 g of fresh weight) (*p* < 0.05). The proportion of MUFA (41.5%) and PUFA (35.5%) in Gemfish roe was higher than in Hoki roe (*p* < 0.05). Generally, palmitic acid (C16:0), oleic acid (C18:1 cn-9), and DHA (C22:6 n-3) were found to be the dominant SFA, MFA, and PUFA, respectively. The fatty acid profile was similar to those reported in King Salmon and Hoki roe [16,20]. The palmitic acid (C16:0), oleic acid (C18:1 cn-9), and docosahexaenoic acid (DHA) (C22:6 n-3) contents in Hoki roe were 14.8%, 23.7%, and 18.6%, respectively and for the Gemfish roe were 12.5%, 29.4%, and 21.4%, respectively. Hoki roe has a slightly lower DHA content compared with previous studies [21], and the difference may be due to differences in maturity, and roe size [22,23]. 

Hoki and Gemfish roes were rich in n-3 fatty acids like eicosapentaenoic acid (C20:5 n-3, EPA), docosapentaenoic acid (C22:5 n-3, DPA), and docosahexaenoic acid (C22:6 n-3, DHA). Gemfish roe extracted oil has a higher (*p* < 0.05) n-3 fatty acid content than Hoki roe, including higher levels of DHA (21.4%), DPA (4.3%), and EPA (5.7%). These results suggest that Gemfish roe offers nutritional value on par or better than Hoki roe, highlighting the potential of both roe types in the development of premium roe products. The n-6/n-3 ratio was <1 for both Hoki and Gemfish roes, which means Hoki roe and Gemfish roe are good sources of n-3 long-chain polyunsaturated fatty acids (LCPUFA) [21].

### 2.3. Phospholipid Composition

The phospholipid content of the percentage of each fatty acid in total fatty acid in Hoki and Gemfish roes is shown in Table 2. The PC content in both Hoki roe and Gemfish roe exhibited the highest content of the total phospholipid, 28.7% and 34.6%, respectively, which is similar to published literature on fish roe that reported PC to be the major phospholipid class [24,25]. PC is produced by the CDP-choline pathway, which is the reason for the high content of the PC in biological samples, compared to other phospholipids. PC can act as a precursor for the biosynthesis of many other lipids and CL can be involved in the mitochondrial energy supply for metabolism [26]. LPS and LDPG were the second and third-highest phospholipids in the samples. PS can be involved in myelin sheath and vital activities of the central nervous system [27]. The two roe samples analyzed in the present study had CL (2.6% for Hoki roe and 3.4% for Gemfish roe) and PS (3.4% for Hoki roe and 1.9% for Gemfish roe). The PI percentage in Hoki and Gemfish roe oil was 4.05% and 3.63%, respectively. The PG content of phospholipid in Hoki roe (6.1%) was significantly (*p* < 0.05) higher than that in Gemfish roe (4.8%). There was no significant difference (*p* > 0.05) in the content of other phospholipids of lipids from the two species.

The PC content in Hoki and Gemfish roe homogenate on µM/100 g wet tissue basis was 231 and 313 µM/100 g roe homogenate, respectively. LPS and LDPG were the second and third highest contents in the two groups of roe oil. The PI content in Hoki and Gemfish roe oil were 34.1 and 33.52 µM/100 g wet tissue, respectively. The content of CL and PC (µM/100 g wet tissue) in phospholipid was significantly (*p* < 0.05) lower in the Hoki roe than in the Gemfish roe group. The LPE was not detected. There were no significant differences (*p* > 0.05) in the PG, LPS, SM, LysoPG, PS and PI contents between the two roe groups.

### 2.4. Enzyme Activity Assessment Using Casein as a substrate

Enzyme hydrolysis of casein was carried out to benchmark the enzymes’ activities. The DH was determined based on tyrosine equivalent at different enzyme concentrations (Alcalase, HT, and FP-II) using casein as a substrate as shown in Appendix A. The results indicated that enzyme concentrations (up to 1 mg/mL) for Alcalase, HT, and FP-II (Appendix A, S1b and S1c, respectively) followed zero-order kinetics under the reaction conditions of 45 °C temperature, pH 7.0, and 30 min hydrolysis time. Casein is a protein commonly used as a reference material for analysis of protease activity, and generally the activity of proteases is reported as U/mL of known protease concentration, or as the specific activity µmol/min/g enzyme of protein [28]. Ahmmed et al. [13] reported the specific activity of Alcalase, HT, and FP-II using casein to be 12.6 × 10^5^ ± 0.47 × 10^5^, 4.88 × 10^5^ ± 0.08 × 10^5^, and 4.1 × 10^5^ ± 0.15 × 10^5^ µmol/min/g enzyme of protein. In the present study, the enzyme activity in relation to hydrolysis of casein was determined to obtain some preliminary information about the relative activity of Alcalase, HT and FPII that could be considered in relation to hydrolysis of Hoki and Gemfish roes homogenates, albeit that roe homogenate is quite a different material compared to casein.

### 2.5. Enzyme Activity on Hoki Roe and Gemfish Roe

The specific activity of the three proteases (Alcalase, HT, and FP-II) on Hoki and Gemfish roe homogenates is shown in Table 3. Alcalase was found to have the highest specific activity compared to HT and FP-II, and HT had a higher specific activity than FP-II (*p* < 0.05) in both Hoki roe and Gemfish roe homogenates (Table 3). This result is in agreement with Ahmmed et al. [13] who reported that the specific activity of Alcalase, HT, and FP-II are 10.5 × 10^5^ ± 0.47 × 10^5^, 4.8 × 10^5^ ± 0.08 × 10^5^, and 3.7 × 10^5^ ± 0.53 × 10^5^ (µmol/min/g enzyme protein), respectively, using Hoki roe homogenate as the substrate. The study of Ryder et al. [14] reported that the specific activity of protease HT and FP-II were 4.4 × 10^6^ ± 2.1 × 10^5^ (fluorescence/min/mg protein) and 2.4 × 10^6^ ± 6.0 × 10^4^ (fluorescence/min/mg protein), respectively, using casein as a substrate, which is the same trend observed for the activity of HT and FP-II found in the present study, but roughly lower one order of magnitude that may be related to batch variation or storage effect. The higher activity of Alcalase compared to HT, and FP-II may be explained by the fact that Alcalase primarily consists of endo-proteases [29], while HT and FP-II contain a mixture of endo- and exo-proteases [14]. 

Endo-proteases cleave peptide bonds within the middle of the protein, while exo-proteases sequentially cleave amino acids from a protein N- or C-termini. Hence endo-proteases typically fragment proteins into peptides, whereas exo-proteases release individual amino acids [30]. Overall, Alcalase, HT, and FP-II exhibited higher activities using Hoki roe homogenate as a substrate than Gemfish roe homogenate (*p* < 0.05). Fish roe is a complex natural product containing many components, including protein, lipid, and cholesterol that could affect the hydrolytic activity of proteases and kinetics of hydrolysis reactions [1].

### 2.6. Determination of Degree of Hydrolysis (DH)

Alcalase is commercially supplied as a solution whereas the HT and FP-II proteases are commercially supplied as powders. Hence in the present study, the hydrolysis experiments were conducted using % *v*/*w* (volume of protease to wet weight of roe sample) for Alcalase and using % *w*/*w* (weight of protease to wet weight of roe sample) for the HT and FPII proteases, to achieve specific information in relation to the commercial batches of protease used and their effect on hydrolysis of the roe. All three proteases have relatively broad pH and temperature optima [13]. Hence to limit the number of variables in the study, a pH of 7.5 and a temperature of 45 °C were used, based on that reported previously [13]. The results from the DH of all treatments show that higher hydrolysis rates were found at the initial hydrolysis stage that gradually decreased over the time course. This is due to a reduction in substrate availability or the generation of inhibitors [31]. Substantial hydrolysis was achieved within 8 h for Hoki roe and 24 h for gemfish roe, beyond which a plateau was achieved. In the initial stage of hydrolysis, the homogenate contains a higher amount of protein substrate than in subsequent stages. During this phase, susceptible peptide bonds are preferentially hydrolyzed, and the hydrolysis products start to occupy binding sites that have unhydrolyzed peptide bonds, which leads to a reduction in the rate of hydrolysis [32,33]. The concentration of the substrate and enzyme is a key factor affecting both the DH and the time required to achieve complete hydrolysis [34]. Ahmmed et al. [13] reported a complete hydrolysis of Hoki roe homogenate by Alcalase, FP-II, and HT within 3 h. The different enzymatic activity in the current study may be caused by the storage time (the HT and FPII were stored at −80 °C whereas Alcalase was stored at 5 °C as per manufacturer recommendation) and proteases may be auto-hydrolyzed.

The roe protein hydrolysis of Hoki and Gemfish in different treatments is shown in Figure 1, Figure 2, Figure 3 and Figure 4. The observed variation in DH can be attributed to the differing protein and lipid composition present in the roe of the two species. In the study by Vlieg and Body [35] that investigated the proximate composition of roes of several New Zealand commercial fish species, it was apparent that the proximate composition of most fish roe differs among the species and across different seasons. There has been no study reporting the proximate composition of Gemfish roe, but as shown in Section 3.1, it is evident that there are significant composition differences (*p* < 0.05) between Hoki roe and Gemfish roe. A high protein content means more proteins can be hydrolyzed, which may increase the efficiency of protease action because the protease has more substrate to act on. However, if the protein is overly dense there may be some protein aggregation, which may also reduce enzyme activity due to substrate inhibition [36]. The lipid of the fish roe potentially can affect the accessibility of proteases to proteins. Fish roe with a high lipid content may be less efficiently hydrolyzed because of the hindering effect of the fat molecules, which limit the protease access to the protein. Fatty acids may bind to proteins and alter their three-dimensional structure, which affects the proteases’ interaction with proteins [37,38]. Another reason that can explain the observed differences in the hydrolysis capacity of the three proteases is their hydrolytic specificity. The protease source and the substrate used for the hydrolysis process can impact the rate and the extent of the hydrolysis process. For example, Kumar et al. [39] obtained trypsin from three different fish *Catla catla*, *Labeo rohit*, *Hypophthalmichthys molitrix* were used to hydrolysis casein, soybean meal, silver carp, and Chilean fish meal and the result demonstrated that the DH is affected by the enzyme source and substrate composition [39].

#### 2.6.1. Fresh Roe Homogenate without Delipidation

The data obtained from the time course hydrolysis of Hoki and Gemfish roe homogenates (Figure 1) show progressive hydrolysis by the three proteases used in the study as shown by the increase in L-serine equivalent contents over the hydrolysis time. For Hoki roe, Alcalase exhibited an overall higher DH compared to HT and FP-II, and the hydrolysis with FP-II was lower than that of HT. These results are in agreement with the findings of Ahmmed et al. [13] who reported Alcalase to have a better hydrolysis capacity with Hoki roe homogenate than HT and FP-II, under the same hydrolysis conditions, and that the DH is increased with increasing the protease concentration. Similar results have been reported by Noman et al. [40] that demonstrated the DH of fresh Chinese Sturgeon (*Acipenser sinensis*) (pH 8.5, time 6 h) was increased with increasing Alcalase enzyme: substrate ratio. The hydrolysis of Gemfish roe was similar for all proteases, but FP-II hydrolysis exhibited more rapid hydrolysis early in the hydrolysis time course compared to Alcalase and HT. 

In all the hydrolysis time courses, using a higher amount of protease (2%, 6%, and 10%) resulted in higher hydrolysis (Figure 1). Hydrolysis of Hoki roe with Alcalase and HT indicates that there is only a small difference in the extent of hydrolysis as a function of the amount of protease used, but hydrolysis with FP-II indicates that, although hydrolysis with 2% and 6% protease is similar (*p* > 0.05), the hydrolysis is enhanced at 10% protease treatment level (*p* < 0.05). Hydrolysis of Gemfish roe with Alcalase and HT indicated that there is some difference in the extent of hydrolysis as a function of how much protease is used (*p* < 0.05), which is more marked with the use of FP-II, hydrolysis from 2% to 6% (*p* < 0.05) and hydrolysis from 6% to 10% (*p* < 0.05).

#### 2.6.2. Fresh and Delipidation

The results obtained from the time course hydrolysis of fresh and delipidated Hoki roe and Gemfish roe are shown in Figure 2. The overall DH of fresh-delipidation samples was lower than the fresh sample. The two studies by Ghelichi et al. [8,41] investigated the production of hydrolysates using common carp roe and delipidated common carp roe, respectively. Both studies used Alcalase 2.4 L FG at a concentration of 1.5% (*v*/*w*) to hydrolyze the sample. The hydrolysis time was between 30 to 120 min, at pH 8, and 50 °C. Comparing the results of these two studies revealed that the degree of hydrolysis (DH%) of delipidated carp roe homogenates was lower than that of untreated carp roe homogenates at the various hydrolysis times. This may be caused by a reduced fluidity of substrates and changes in protein three-dimensional structure caused by exposure to organic solvent [42,43]. The delipidated Hoki roe hydrolyzed by Alcalase exhibited an overall higher (*p* < 0.05) DH compared to HT and FP-II. Overall, hydrolysis with FP-II was lower than that of HT and was similar to the fresh roe sample. The extent of Gemfish roe hydrolysis by the three proteases was similar, but FP-II hydrolysis exhibited more rapid hydrolysis early in the hydrolysis time course.

Hydrolysis using different amounts (2%, 6%, and 10%) of protease generally indicated that the extent of hydrolysis increased with the increase of protease amount. This is a general characteristic of enzymatic hydrolysis of proteins [44]. Increasing the enzyme concentration can accelerate peptide bond cleavage and prevent aggregate formation. Hydrolysis of Hoki roe with Alcalase, HT and FP-II stated that there is only a small difference (*p* > 0.05) in the extent of hydrolysis as a function of the amount of protease used. Hydrolysis with 2% and 6% did not exhibit a significant difference (*p* > 0.05), but the hydrolysis was enhanced with the 10% protease treatment. The hydrolysis of Gemfish roe homogenate with Alcalase indicates some differences (*p* < 0.05) in the extent of hydrolysis as a function of protease concentration.

The three enzymes exhibited decreased DH in the two roe homogenates after delipidation compared to non-treated samples, and the hydrolysis achieved equilibrium more rapidly over time. The changed amenability of proteins and other hydrolysable components in the roe homogenate to hydrolysis after defatting could explain this phenomenon. The hydrophilicity of the three enzymes could be a contributing factor.

#### 2.6.3. Freeze-Dried without Delipidation

The results obtained from the time course hydrolysis of freeze-dried and without delipidation Hoki roe homogenate and Gemfish roe homogenates are shown in Figure 3. The overall DH of freeze-dried samples was lower than fresh samples (*p* < 0.05). Similar to fresh and fresh-delipidation treatments, Alcalase exhibited an overall higher DH compared to HT and FP-II (*p* < 0.05). The extent of hydrolysis of Gemfish roe was similar among all three proteases, but FP-II hydrolysis exhibited more rapid hydrolysis kinetics early in the hydrolysis time. Hydrolysis using different amounts of proteases (2%, 6%, and 10%) resulted in an overall hydrolysis increase with the increase in the protease concentration.

Hydrolysis of Hoki roe by Alcalase and FP-II indicated that there is only a small difference in the extent of hydrolysis as a function of the amount of protease used. For Alcalase, the hydrolysis with 2% and 6% did not exhibit a significant difference (*p* > 0.05) but the hydrolysis was enhanced with the 10% of protease treatment. For FP-II, the 2% with 6% hydrolysis treatments and 6% with 10% treatments were not different, but the hydrolysis was significantly enhanced by increasing the protease concentration from 2% to 10%. In the gemfish roe, hydrolysis with the Alcalase, HT, and FP-II indicates a small difference in the extent of hydrolysis as a function of the protease concentration.

However, the extent of hydrolysis of the freeze-dried fish roe homogenates by the three enzymes was lower than that achieved for the roe treatments as discussed above, which may be related to changes in the protein structure caused by the freeze drying, which led to low enzyme-substrate contact efficiency [45]. Dehydration by heating or lyophilization can result in protein denaturation. Such alterations could modify the secondary, tertiary, or quaternary structures of the protein molecules [46]. Indications of protein denaturation are typically observed as modification of their solubility. Córdova-Murueta et al. [47] investigated the effects of different drying processes on the hydrolysis of fish muscle protein. The obtained results indicated that the freeze-drying process decreases the water-soluble protein content compared to the thermal drying process. The freeze-drying process had the highest DH in all tested fish muscles. Fresh roe homogenates may have a looser structure than freeze-dried ones, which may facilitate easier contact and better hydrolysis of the protein by the proteases. It was suggested that freeze-drying might result in the samples becoming more compact or forming aggregates, thereby reducing the efficiency of enzyme-substrate interaction [48].

#### 2.6.4. Freeze-Dried and Delipidation

The data from the time course hydrolysis of freeze-dried and delipidated Hoki and Gemfish roe homogenates are presented in Figure 4. The results show that hydrolysis with all three proteases (Alcalase, HT, and FP-II) progressively hydrolyzed the Hoki and Gemfish roe proteins. Alcalase achieved a higher DH compared to HT and FP-II, with HT exhibiting slightly lower DH than FP-II. In the Gemfish roe samples, all three proteases showed similar hydrolysis patterns, but FP-II had more rapid kinetics early on.

Using higher concentrations of protease (2%, 6%, and 10%) generally increased the extent of hydrolysis. For Hoki roe treated with Alcalase and FP-II, there was little difference between 2% and 6%, but a significant increase was found at 10%. Similarly, increasing the concentration of HT from 2% or 6% to 10% resulted in a significant increase in Hoki roe hydrolysis. In Gemfish roe, all proteases showed small differences in hydrolysis extent with varying amounts. There was no significant difference between 2% and 6%, but a significant increase was observed by increasing the protease concentration from 2% or 6% to 10%.

The results suggest that both delipidation and freeze-drying pretreatments decrease the hydrolysis kinetics. The DH was lower for combined delipidated freeze-dried treatments than for either of the treatments alone. This supports the contention that these pretreatments affect the protein structure and substrate environment, thus reducing hydrolysis more than when a single pretreatment is used.

### 2.7. SDS-PAGE Analysis

The three gels in Figure 5a–c represent the three proteases (Alcalase, HT, and FP-II) control before (Figure 5a) and after (Figure 5b) the 24 h hydrolysis of the 4 treatment groups along with the control. The fish roe proteins under all pretreatment conditions exhibited a distribution of bands with a range of molecular weights from 260 kDa to 3.5 kDa (Figure 5c). Comparing three proteases reveals that FP-II has a subtle self-hydrolysis phenomenon during incubation. The time course (0–24 h) hydrolysis of the Hoki roe and Gemfish roe homogenates of the four different treatments (Fresh, fresh-delipidation, freeze-drying, and freeze-drying with delipidation) hydrolyzed by 10% (*v*/*w* or *w*/*w*) of three proteases (Alcalase, HT, and FP-II) were analyzed by1D-SDS-PAGE (Figure 6a–f).

The two protein bands from 40 kDa to 30 kDa were absent after 24 h incubation at 45 °C. For the rest of the bands, various concentrations of all three proteases retained protein bands corresponding to their original molar masses (Figure 6a,b). The presence of bands at low molecular weight in fish roe before hydrolysis was unexpected. In the report by Ghelichi et al. [8], minor peaks of a relatively low molecular weight range (3–14 kDa) were observed in defatted carp roe homogenate, potentially due to an autolytic process occurring in fish roe and/or the defatting process leading to the formation of low-molecular-weight peptides. The protein band distributions in the homogenates of Hoki roe and Gemfish roe were similar, with Hoki roe exhibiting more protein fragments above 20 kDa compared to Gemfish roe, and after the delipidation there was a protein band absence at 60 kDa in both freeze-dried and fresh Hoki roe homogenate. Before hydrolysis, the protein bands of the non-delipidation fish roe were all distorted, with fresh Hoki roe showing the most prominent distortion. Additionally, the freeze-dried Hoki roe and Gemfish roe samples as well as the fresh Gemfish roe, all exhibited distortion around 3 kDa (Figure 6c). 

The summary protein image of the time-course hydrolysis of Hoki and Gemfish roe homogenate revealed that all three proteases exhibited a non-selective hydrolytic manner, indicating no apparent specificity towards any of the roe proteins (Figure 6). Overall, the results indicated that proteins in both Hoki and Gemfish roe homogenates were hydrolyzed over time by all three proteases and treatments. However, unexpected bands with high molecular weight at 50–40 kDa were present in FD-dL Hoki roe and Gemfish roe samples which was hydrolyzed by 10% FP-II. Among the three proteases, Alcalase was more effective in hydrolyzing Hoki and Gemfish roe homogenates compared to HT and FP-II. After 24 h of hydrolysis, protein bands were absent in Hoki roe hydrolysate treated with Alcalase. HT showed a better hydrolysis effect than FP-II, as evidenced by the presence of protein bands with higher molecular weight in the FP-II image (Figure 6e,f). Ahmmed et al. [13] reported that Alcalase was more effective in hydrolyzing Hoki roe homogenate than HT and FP-II, based on SDS-PAGE analysis. Fresh Hoki roe and Gemfish roe homogenates exhibited better hydrolysis than freeze-dried samples.

Distortion was also observed in samples containing lipids, especially in the Hoki roe homogenate. Pokhariyal et al. [49] reported that the complexity of protein bands in the SDS-PAGE was increased in the presence of lipid, which led to distortion of protein bands. Distinguishing the effect of the delipidation process on protein bands in SDS-PAGE of Hoki and Gemfish roe hydrolysates proved to be quite challenging.

### 2.8. Antioxidant Activity

In the present study, the results of DH and antioxidant activity of Hoki and Gemfish roe homogenates with different treatments, indicate some comparability. In general, higher antioxidant activity was found with higher DH (Figure 7), which is consistent literature [50]. Ghelichi et al. [8] reported that the antioxidant activity (DPPH, ABTS, and FRAP assays) of carp roe hydrolysate produced by Alcalase and reported an increase in the antioxidant activity with increasing the DH of the sample. Figure 7 showed that the antioxidant activity assay (DPPH and ABTS are reported as µmol Trolox equivalent per g of dry hydrolysate material, and the FRAP assay was reported as FeSO_4_·7H_2_O equivalent per gram of freeze-dried hydrolysate sample) was dependent on the enzyme type used in the hydrolysis and the analysis of the antioxidant activity of the Gemfish roe and Hoki roe homogenate.

#### 2.8.1. Antioxidant Activity of Different Enzyme Treatments and Fish Roe

Figure 7I shows the antioxidant activity of Gemfish roe and Hoki roe homogenates treated with different proteases (Alcalase, HT, and FP-II). Alcalase and FPII generated hydrolysates with the highest DPPH˙ scavenging activity in both Hoki and Gemfish roe, which was significantly higher than HT (*p* < 0.05) (Figure 7I(a)). The antioxidant activities were not affected by the fish roe type (*p* > 0.05). The ABTS˙ scavenging activity was not affected (*p* > 0.05) by different proteases or the fish roe type (*p* > 0.05) (Figure 7I(b)). The ferric reducing power was significantly affected (*p* < 0.05) by the protease type in both Hoki and Gemfish roes (Figure 7I(c)), but not affected by the roe type (*p* > 0.05). Furthermore, the ferric-reducing power of Alcalase and HT hydrolysates was significantly higher than FP-II (*p* < 0.05), but the ferric-reducing power of Alcalase and FP-II hydrolysates was not different (*p* > 0.05) in both roe species samples. Rajabzadeh et al. [6] compared Alcalase and pepsin abilities in hydrolyzing rainbow trout protein and found higher antioxidant activity in Alcalase-treated samples. Furthermore, Ahmmed et al. [13] demonstrated that Alcalase produced the highest antioxidant activity and DH when used to hydrolyze Hoki roe protein, compared to HT and FP-II. While HT and FP-II are not commonly used for fish roe hydrolysis, they were successful in hydrolyzing meat proteins, with HT being particularly effective for myofibrils and connective tissues [14]. Ryder et al. [15] found that HT hydrolysates exhibited high antioxidant activity. This suggests that, while Alcalase generally performs better, HT and FP-II also yield substantial antioxidant activity since no difference in their antioxidant activity was detected among these three enzyme hydrolysates in ABTS and FRAP tests of Hoki roe.

#### 2.8.2. Antioxidant Activity of Fish Roe Homogenates from Fresh, Fresh-Delipidated, Freeze-Dried and Freeze-Dried-Delipidated Treatments

Figure 7II shows the antioxidant activity of Gemfish roe and Hoki roe hydrolysates from different treatments [fresh (F), fresh-delipidated (dL), freeze-dried (FD), freeze-dried-delipidated (FD-dL)]. Fresh roe had the highest DPPH˙ scavenging activity in both Hoki and Gemfish roe homogenates (*p* < 0.05). There was no significant difference in the DPPH˙ scavenging activity dL and FD treatments (*p* > 0.05) in both roe types. The lower antioxidant activity of F treatment compared to the dL may be due to the loss of peptides during the delipidation process [21]. The loss of peptides during the delipidation process was also reported for protein hydrolysates obtained by hydrolyzing mung bean protein for 120 min using Alcalase [51]. Furthermore, it is known that certain lipids/fatty acids can act as antioxidants, such as polyunsaturated fatty acids (PUFAs) and certain phospholipids [52,53] and their removal will reduce the total antioxidant activity. For instance, Zhu et al. [54] reported phospholipids from Large yellow croaker roe exhibited substantial antioxidant activity. The FD-dL samples had the lowest DPPH˙ scavenging activity in both Hoki and Gemfish roe samples. This is supported by the findings of Vu et al. [55] who reported higher DPPH˙ scavenging activity and FRAP activity in untreated Red Sea cucumber (*Parastichopus tremulus*) protein extracts compared to their freeze-dried counterparts. The effects of different pretreatments on the ABTS˙ scavenging activity of fish roe hydrolysates were similar to those observed in the DPPH assay (Figure 7III(b)). The ferric reducing power of Hoki roe hydrolysates was affected by different treatments and was similar to the DPPH and ABTS assays. The Gemfish roe F treatment had the highest (*p* < 0.05) ferric reducing power, and the activity of the other three treatments did not differ from each other (Figure 7II(c)).

#### 2.8.3. Antioxidant Activity of Different Treatments and Enzyme Treatments

Figure 7III shows the hydrolysates’ antioxidant activities were affected by the different treatments (F, dL, FD, and dL-FD) and enzyme treatments (Alcalase, HT, and FP-II). Roe protein hydrolysates were influenced by enzyme type and fish roe species (Figure 7III(a)). Alcalase had the highest (*p* < 0.05) DPPH˙ scavenging activity of all the treatments. FP-II hydrolysate had higher (*p* < 0.05) DPPH˙ scavenging activity than HT under F treatment. The F treatment had the highest (*p* < 0.05) DPPH˙ scavenging activity and the dL treatments had significantly lower antioxidant activity than F (*p* < 0.05). For Alcalase, the DPPH˙ scavenging activity was significantly higher than FD-dL, and there was no difference among the F, dL, and FD treatments. For HT-treated samples, the DPPH˙ scavenging activity was not affected by the different lipid removal and freeze-drying treatments (*p* < 0.05). The DPPH˙ scavenging activity of the FP-II-treated samples was not different among the dL, FD, and dL-FD treatments. There was no effect for the different proteases on the ABTS˙ scavenging activities of the F, dL, and FD-dL treatments (*p* > 0.05) (Figure 7III(b)). Generally, FD treated with Alcalase had higher (*p* < 0.05) ABTS radical scavenging activity than HT and FP-II, and the ABTS radical scavenging activity of HT and FP-II did not show any significant differences (*p* < 0.05). The F treatment had the highest (*p* < 0.05) ABTS radical scavenging activity. The ABTS˙ scavenging activity of Alcalase-treated samples showed a significantly higher activity in dL treatment compared to dL-FD, and there was no difference between dL with FD-dL. For HT, ABTS˙ scavenging activity effect by dL was significantly higher than FD and FD-dL. There was no difference between F with dL, and FD with FD-dL. For FP-II, the ABTS˙ scavenging activity had no significant difference (*p* > 0.05) between dL, FD, and dL-FD. For FP-II the effect of ABTS˙ scavenging was similar to the FP-II result in the DPPH method (Figure 7III(a)). For the FRAP assay, the result of Ferric (Fe^3+^) reducing power effect by different treatments and different proteases was similar to the ABTS assay (Figure 7II(c)). The difference was the Ferric (Fe^3+^) reducing power of HT hydrolysate was significantly higher than HT under the effect of FD treatment. The present study indicates that freeze-drying with delipidation roe hydrolysate samples results in lower antioxidant activity than delipidation and freeze-drying with lipids, and this indicates that the two treatment methods act separately on the antioxidant activity of the fish roe hydrolysates.

## 3. Materials and Methods

### 3.1. Chemicals and Reagents

Commercial Alcalase was from Sigma (Sigma-Aldrich, Auckland, New Zealand). FP-II and HT protease preparation powders were supplied by McConnell Bros. (formerly Enzyme Solutions Pty. Ltd.; Croydon South, Victoria, Australia). HCl (37%, Lot# Z0701617051) (Supelco Ltd., Burlington, MA, USA). Iron chloride hexahydrate (FeCl_2_·4 H_2_O) (BDH Chemicals Ltd., Lot#6343980, Poole, England). Iron sulfate heptahydrate (FeSO_4_·7H_2_O) (Merck Ltd., Rahway, NJ, USA). Invitrogen BoltTM 4–12% Bis-Tris Plus gels (Ref: NW04122BOX; Lot: 22081710), Invitrogen SimplyBlueTM SafeStain (Ref: 46-5034; Lot: 2474562), Invitrogen BoltTM MES SDS Running Buffer (20x) (Lot# 2505599), Novex^®^ Sharp Pre-Stained Protein Standard. Casein, L-tyrosine (Lot# T-3754), 2,2-dipheny1-1-picrylhydrazyl (DPPH), 6-hydroxy-2,5,7,8-tetramethyl chroman-2-carboxylic acid (Trolox)(Lot# 01401HU), 2,2′-azino-bis(3-ethylbenzothiazoline-6-sulfonic acid (ABTS) (Lot#SLBJ5208V), APS, 2,4,6-Tris(2-pyridyl)-s-triazine (TPTZ) (Lot# BCBK6346V) were from Sigma-Aldrich Ltd. (St Louis, MO, USA).

### 3.2. Sample and Protease Preparations

Frozen Hoki roe and Gemfish roe (grade 4, 20 kg each) were obtained from a commercial seafood company in New Zealand. The roe was thawed overnight at 4 °C, cut into small pieces, and homogenized in a blender (Watson Victor Limited, Wellington, New Zealand). The homogenized roe samples were passed through a sieve to remove connective tissue and blood vessels from the roe sample. The homogenized samples were aliquoted and separately stored at −20 °C for further use. Both Hoki and Gemfish roe homogenate were subjected to four different treatments (fresh, fresh with delipidation, freeze-drying, and freeze-drying with delipidation) and were used as substrates to investigate the hydrolysis capacity of the proteases Alcalase, HT, and FP-II.

### 3.3. Proximate Composition

The proximate analysis of Hoki and Gemfish roe samples was conducted as described by Ahmmed et al. [20]. All samples were analyzed using individual roes (*n* = 3). Moisture content was determined gravimetrically, and moisture content was quantified by measuring the weight loss before and after drying. Total protein content was measured using the Kjeldahl method (Kjeltec9 SOP system, FOSS Tecator AB, Högänas, Sweden). Total ash content was determined using a muffle furnace at 550 °C. Lipid content was determined using the ETHEX method described by Ahmmed et al. [20]. Total carbohydrate content was calculated using the Equation (1).
% of total carbohydrate content = (100 − sum (moisture + lipid + ash + protein) contents) %(1)

### 3.4. Fatty Acid Analysis by GC-FID

The fatty acids present in the lipids of fish roe were prepared as mentioned in Section 3.3 and analyzed as described by Ahmmed et al. [20,56]. A 100 mg of the extracted oil sample was dissolved in 1 mL of methanol: chloroform (2:1, *v*/*v*), then an internal standard (C13:0, 1 mg/10 mL lipid) was added to the sample, and 400 µL of the sample was mixed with 3 mL KOH and transferred into a long neck glass Kimax tube and placed in a heating block at 80 °C for 20 min. Then was cooled, 2 mL diethyl ether and 5 mL deionized water were added and mixed by vortex. The solution was then allowed to phase separate, and the upper layer was discarded. Four drops of 37% (*v*/*v*) HCl and 2 mL diethyl ether were added to the lower phase, vortexed, and then the solution was allowed to phase separate the upper layer was then transferred to a new glass Kimax tube and 2 mL 14% *w*/*v* BFX3 was added and mixed, and the mixture was heated at 80 °C for 20 min. After the tube was cooled down, 5 mL of saturated NaCl solution was added and vortexed. Then, the solution was allowed to phase separate, and the upper layer containing the FAMEs transferred into a 2 mL GC vial for analysis by gas chromatography with flame ionization detection (GC-FID, Agilent 6890N, Agilent Technologies Inc., Santa Clara, CA, USA).

H_2_ was used as a carrier gas at a flow rate of 1 mL/min with a split ratio of 30:1. A 50-m GC column (BPX-70 silica column (Phenomenex, Torrance, CA, USA), 0.32 mm inner diameter, with a 0.25 µm film thickness was used. The GC operating parameters were as follows: initial column temperature 100 °C, then increased at 10 °C /min to 160 °C, and then at a 3 °C /min increase to 220 °C, which was maintained for 5 min and then raised at 10 °C /min to 260 °C and held for 5 min. Individual fatty acids were identified and quantified based on retention time and peak area, respectively, of commercial FAME standards (FAMQ-005; AccuStandard Inc., NH, USA) as shown in Appendix A. Peaks that could not be identified based on the retention time of the FAME standards were reported unknown.

### 3.5. Phospholipid Analysis by 31P NMR

A 150 mg oil sample was dissolved in 1 mL aqueous detergent solution [10% (*w*/*v*) NaCl, 1% (*w*/*v*) EDTA, 20% (*v*/*v*) D2O, 0.83 µmol/mL glyphosate], and the pH was adjusted to 7.4 with 6 M NaOH. The solution was vortexed and placed in an ultrasonic bath (Elmasonic Dentec, Auckland, New Zealand) at 60 °C for 10 min. The sample was then centrifuged at 21,000× *g* for 10 min using an IEC Micromax centrifuge (DJB Labcare limited, Buckinghamshire, UK), and the bottom transparent phase (800 µL) was transferred into a 5 mm NMR tube for analysis.

The 31P NMR analysis used a Varian MR 400 NMR (Agilent Technologies, Santa Clara, CA, USA). The operating conditions of the NMR instrument were: 25 °C, excitation pulse 90 °C, sweep width 6067 Hz, collecting 65536 data points. Glyphosate in D2O was used as an internal standard for quantification of the individual phospholipids in the fish roe oil. Nine commercial standards determined their chemical shifts (Appendix A). The chemical shifts for other phospholipids in both solvent systems were referenced from Ahmmed et al. [13]. NMR results were processed and analyzed using the Mestrenova system (version 12.0.3, Mestrelab Research, Coruña, Spain).

### 3.6. Enzyme Activity Assessment Using Casein, Hoki, and Gemfish Roe Homogenates as Substrates

The enzyme activity assessment described by Ahmmed et al. [13]. The result is reported as the Equation (2).
(2)$$\mathrm{Units}/\mathrm{mL}\;\mathrm{protease}=\frac{umol\;tyrosin\;equivalents\;\times \;VT}{VE\times T\times VA}$$

VT: Total volume (in mL) of assay

VE: Volume of protease (mL)

T: Time of assay (min) as per unit definition

VA: Volume (mL) used in microplate reader.

### 3.7. Determination of Degree of Hydrolysis

The degree of the hydrolysis (DH) analysis was as described by Ahmmed et al. [13]. The protein concentration of the Hoki roe and Gemfish roe homogenates was adjusted to 6.5 mg/mL and the samples were hydrolyzed with the proteases (Alcalase, HT and FPII) at 2%, 6%, and 10% (*v*/*w* or *w*/*v*). Detailed information on these treatments is provided in Appendix A.

Four roe homogenate sample treatments (Fresh, fresh-delipidation, freeze-drying, and freeze-drying with delipidation), two fish roe species (Gemfish roe and Hoki roe), three proteases (Alcalase, HT, and FP-II), and 3 protease concentrations (2%, 6%, and 10%) were formulated to generate a total of 72 treatment hydrolysis groups that were analyzed in triplicate samples.

All samples were hydrolyzed at 45 °C and pH 7.5 (optimized according to the study presented by Ahmmed et al. [13]). The pH of the samples was adjusted using 0.1 M HCl or NaOH, and samples were preheated in an incubator at 50 °C for 10 min. The protease was solubilized in sodium acetate buffer (0.1 M, pH 7), and 1 mL of the protease solution was added to each separate roe sample, which was vortexed for 5 s before hydrolysis. Time-course aliquots (600 µL) were collected at 0, 1, 2, 3, 4, 6, 8, 24, and 35 h to 1.7 mL centrifuge tubes, and 500 µL of 20% trichloroacetic acid (TCA) was added to inactivate the protease. The samples were centrifuged at 21,000× *g* for 20 min, and the supernatant was collected and transferred to another 1.7 mL microcentrifuge tube. The samples were then centrifuged again at 21,000× *g* for 10 min to obtain the final supernatant.

The DH was determined using the OPA method [13]. L-serine standard 0–0.2 mg/mL was generated was used to construct a standard curve. and the result was reported as mg L-serine equivalent.

### 3.8. SDS-PAGE Protein Profile of Hoki and Gemfish Roe Hydrolysates

Hoki roe and Gemfish roe hydrolysate samples at time points 0 h, 1 h and 24 h were analyzed by SDS-PAGE. Protein samples (10 µL) were mixed with 10 µL Type 1 MilliQ water, 7.6 µL Novex™ BOLT™ LDS sample buffer (4×, Life Technologies, Aukland, New Zealand, and 1.5 µL of 10×reducing agent in a 0.5 mL centrifuge tube. The samples were mixed and incubated at 70 °C for 5 min in a programmable 480 Cetus DNA Thermal Cycler (Perkin Elmer Inc., Waltham, MA, USA.) and then cooled to room temperature. The prepared samples were loaded onto a BOLT™ 4–12% Bis-Tris 1.0 mm mini protein gel. A protein standard was also loaded in one lane of the gel. Electrophoresis was performed in BOLT™ MES SDS running buffer (20×, Life Technology) at a constant 165 V and 120 mA for 35 min at 22 °C. The gel was rinsed four times in distilled water for 10 min each, then stained overnight in 20 mL of SimplyBlue™ SafeStain with gentle shaking. The gel was then destained in Type 1 Milli-Q water, and the gel image was captured using a CanoScan LiDE 600F scanner (Canon Inc., Auckland, New Zealand).

### 3.9. Antioxidant Activity

The antioxidant activity assays used in this study were the DPPH, ABTS, and FRAP methods as reported by Brand-Williams et al., Re et al., and Benzie et al. [57,58,59]. A stock DPPH solution (0.24 mg/mL) was prepared in absolute ethanol and its absorbance was adjusted to 1 at 517 nm. The assay was carried out at an absorbance of 517 nm and 37 °C using a 96-well microplate covered with a preservative film to prevent evaporation. A Trolox standard curve was constructed in a concentration range of 0 to 200 µg/mL. The incubation time for the DPPH assay was 8 h, and the DPPH˙ scavenging activity is reported as µmol Trolox equivalents per gram of sample.

For the ABTS assay, a 10 mL 7 mM ABTS stock solution and 1 mL of 254 mM APS stock solutions were prepared fresh, and both were kept out of direct light. After adding 100 µL APS to the 10 mL ABTS stock, the mixture was incubated overnight to generate the ABTS free radical. A 96-well microplate was set up in a plate reader at a temperature of 37 °C and an absorbance of 734 nm. The absorbance of the ABTS stock solution was adjusted to 0.7 with deionized water. A Trolox standard curve was created with concentrations from 0 to 400 µg/mL. The incubation time for the ABTS assay was 2.5 h, and results were reported as the ABTS˙ scavenging activity in µmol Trolox equivalents per gram of sample.

For the FRAP assay, a 10 mL of 10 mM TPTZ in 40 mM HCl solution and a 20 mM FeCl_3_·6 H_2_O solution were prepared. A FeSO_4_·7H_2_O standard was made with concentrations ranging from 0 to 2000 µM. The incubation time for the FRAP assay was 6 h. Results were reported as the ferric reducing antioxidant power in µM FeSO_4_·7H_2_O equivalents per gram of sample.

### 3.10. Statistical Analysis

All of the roe hydrolysate samples were prepared in 4 replicate samples ha. The data were reported as the mean ± standard deviation (SD) of three individual replicate roes. A one-way analysis of variance (ANOVA) was used to assess the effects of four treatments (fresh, fresh-delipidation, freeze-drying, and freeze-drying with delipidation) on DH and antioxidant activity, using Minitab^®^ version 15.1.0 (Minitab Limited, Sydney, Australia). Tukey’s test at the 0.05 significance level was used to identify statistically significant differences among the means.

## 4. Conclusions

This study investigated the nutrient content of Hoki and Gemfish roes and the effects of defatting and freeze-drying treatments on the hydrolysis of their homogenates as fresh, defatted, freeze-dried, and freeze-dried plus defatted. The hydrolysis was carried by three commercial proteases (Alcalase, HT, and FP-II). The low-grade fish roes have substantial contents of nutritional lipids and substantial amounts of protein. The hydrolysis of the roe homogenates at pH 7.0 and 45 °C exhibited that all three protease preparations could reach a maximal DH within 8 h under the conditions used. Alcalase hydrolysates displayed the best antioxidant properties in DPPH, ABTS, and FRAP antioxidant activities. Concurrently, under the same conditions, the DH and antioxidant properties of the two types of roe proteins were reduced with the inclusion of defatting and freeze-drying treatments, potentially due to changes in either protein morphology or loss of antioxidant peptides during lipid extraction, and this requires further research.

## Figures and Tables

**Figure 1 marinedrugs-22-00364-f001:**
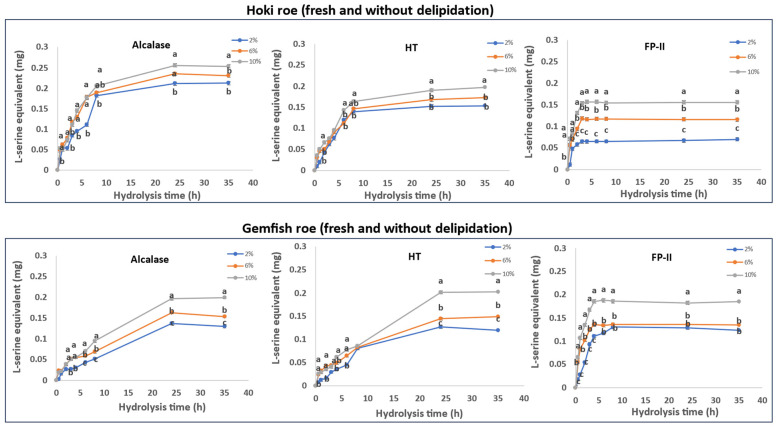
Time course hydrolysis of frozen-thawed Hoki and Gemfish roes with lipid present. Homogenates of fresh Hoki and Gemfish roe were prepared by adding 7.3 g and 5.5 g of roe to 200 mL potassium phosphate buffer (pH 7) to achieve 6.5 mg/mL protein concentration, and aliquots of 20 mL were subject to hydrolysis with three different concentrations (2%, 6%, and 10%) of either microbial protease Alcalase (*v*/*w*), bacterial protease HT (*w*/*w*), or fungal protease FP-II (*w*/*w*), respectively, at 45 °C. Alcalase is commercially available as a solution, and HT and FP-II are commercially available as powders. The degree of hydrolysis was determined based on the L-serine equivalent. The data were obtained from three independent hydrolyses for each roe and protease combination. Analysis of variance (ANOVA) was carried out. Letters a–c indicate significant differences among different samples prepared with varying enzyme concentrations at the same time point. (*p* < 0.05).

**Figure 2 marinedrugs-22-00364-f002:**
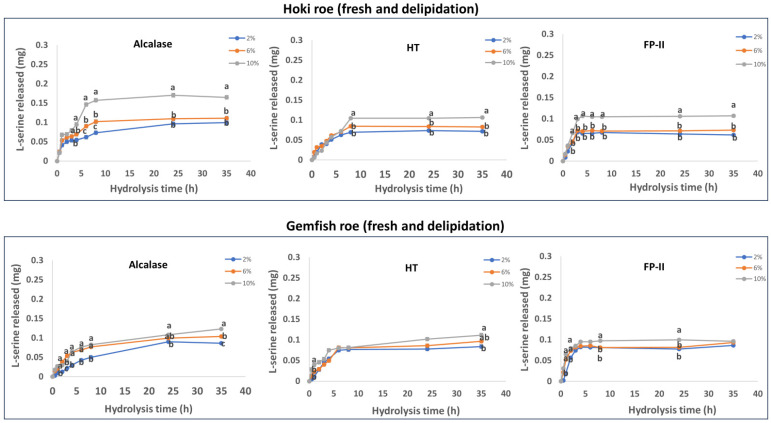
Time course hydrolysis of fresh Hoki and Gemfish roes after removing the lipid fraction. Before hydrolysis, the Hoki roe and Gemfish roe homogenates were delipidated by ethanol and hexane using the ETHEX lipid extraction method. For hydrolysis, homogenates of delipidated frozen-thawed Hoki and Gemfish roes were prepared by adding 7.3 g and 5.5 g of thawed delipidation roe to 200 mL of potassium phosphate buffer (pH 7) to achieve 6.5 mg/mL protein concentration and aliquots of 20 mL were subject to hydrolysis at three different concentrations (2%, 6%, and 10%) of either Alcalase (*v/w*), bacterial protease HT (*w/w*), or fungal protease FP-II(*w/w*) at 45 °C. The degree of hydrolysis of the samples was determined based on the L-serine equivalent method. The data were obtained from three independent hydrolyses for each roe and protease combination. Analysis of variance (ANOVA) was carried out. Letters a–c indicate significant differences among different samples prepared with varying enzyme concentrations at the same time point (*p* < 0.05).

**Figure 3 marinedrugs-22-00364-f003:**
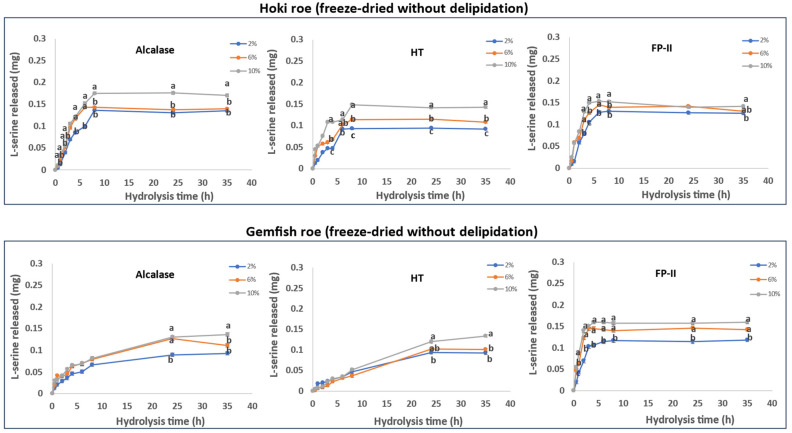
Time course hydrolysis of freeze-dried Hoki and Gemfish roe without lipid extraction. Before hydrolysis, the Hoki roe and Gemfish roe homogenates were freeze-dried. For hydrolysis, homogenates of freeze-dried Hoki and Gemfish roe were prepared by adding 2.76 g and 1.96 g of freeze-dried roe powder to 200 mL of potassium phosphate buffer (pH 7) to achieve the protein concentration 6.5 mg/mL, and aliquots of 20 mL were subjected to hydrolysis with three different concentrations (2%, 6%, and 10%) of either Alcalase (v/w), bacterial protease HT (*w/w*), or fungal protease FP-II (*w/w*) at 45 °C. The data were obtained from three independent hydrolyses for each roe and protease combination. Analysis of variance (ANOVA) was carried out. Letters a–c indicate significant differences among different samples prepared with varying enzyme concentrations at the same time point (*p* < 0.05).

**Figure 4 marinedrugs-22-00364-f004:**
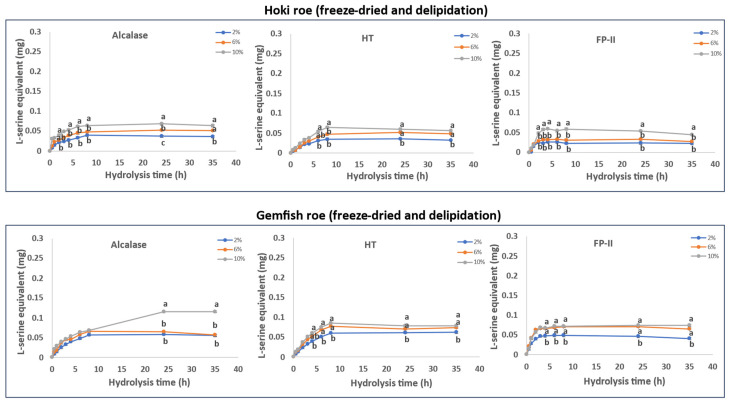
Time course hydrolysis of freeze-dried Hoki and Gemfish roe without lipid present. Before hydrolysis, Hoki roe and Gemfish roe homogenate delipidated and freeze-dried by the ETHEX method and the freeze drier. For hydrolysis, homogenates of freeze-dried Hoki and Gemfish roe were prepared by adding 2.07 g and 2.12 g of delipidated freeze-dried roe powder to 200 mL of potassium phosphate buffer (pH 7), and aliquots of 20 mL were subject to hydrolysis with three different concentrations (2%, 6%, and 10%) of either plant-based protease Alcalase (*v/w*), bacterial protease HT (*w/w*), or fungal protease FP-II (*w/w*), respectively, at 45 °C. Analysis of variance (ANOVA) was carried out. Letters a–c indicate significant differences among different samples prepared with varying enzyme concentrations at the same time point (*p* < 0.05).

**Figure 5 marinedrugs-22-00364-f005:**
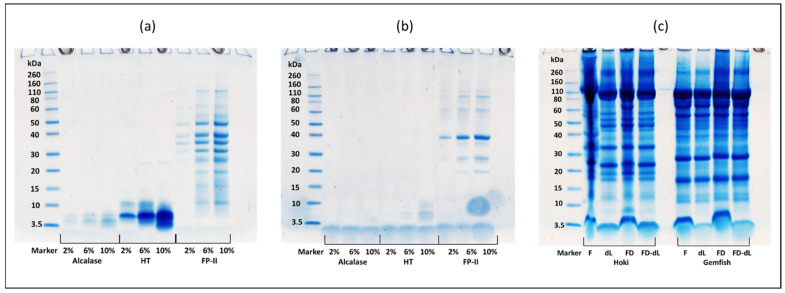
Time course hydrolysis of freeze-dried Hoki and Gemfish roe without lipid present. 1D SDS-PAGE of protein hydrolysis profiles of three proteases (Alcalase, HT, and FP-II) with the concentration [2%, 6%, and 10% (*w*/*v* or *v*/*v*)] control before 24 h incubation (**a**), control after 24 h incubation (**b**), and the sample control of Hoki and Gemfish roe homogenate with 4 treatments before hydrolysis (**c**). F = fresh, dL = delipidation, FD = freeze-dried, and FD-dL = freeze-dried with delipidation. A darker blue band indicates a higher concentration of protein.

**Figure 6 marinedrugs-22-00364-f006:**
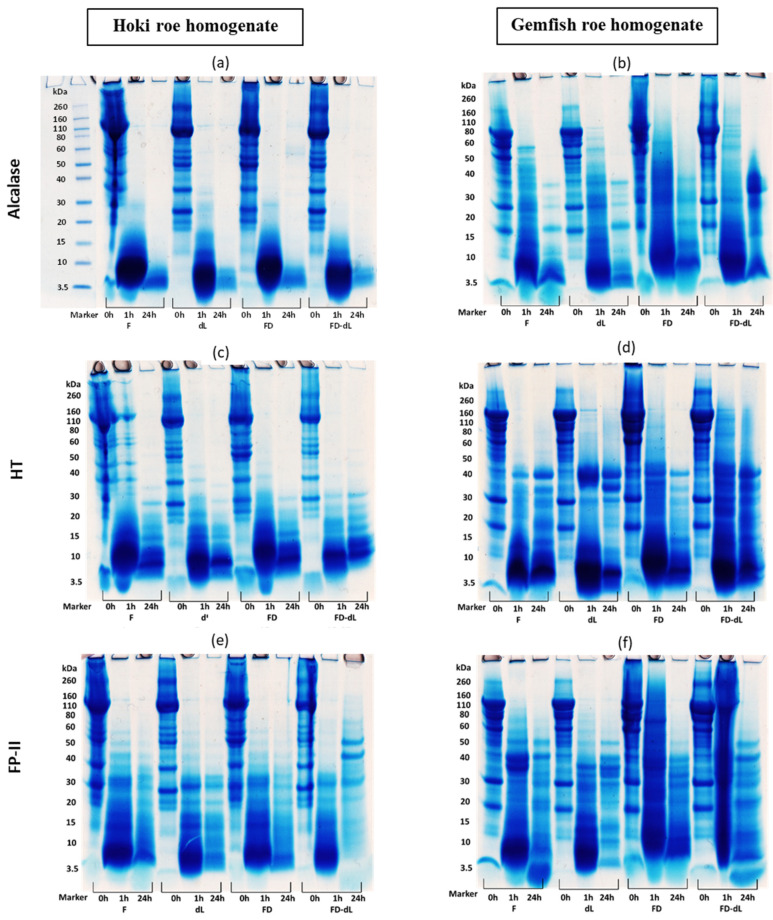
1D SDS-PAGE of protein hydrolysis profiles of Hoki roe (**a**,**c**,**e**) and Gemfish roe (**b**,**d**,**f**) homogenate. Alcalase (**a**,**b**), HT (**c**,**d**), and FP-II (**e**,**f**) all with the same amount of protease concentration (10% *v*/*w* or *w*/*w*) and effect of 4 treatments (F, dL, FD, and dL-FD) on protein hydrolysis of Hoki and Gemfish roe homogenate. F = fresh, dL = delipidation, FD = freeze-dried, and FD-dL: freeze-dried with delipidation. A darker blue band indicates a higher concentration of protein.

**Figure 7 marinedrugs-22-00364-f007:**
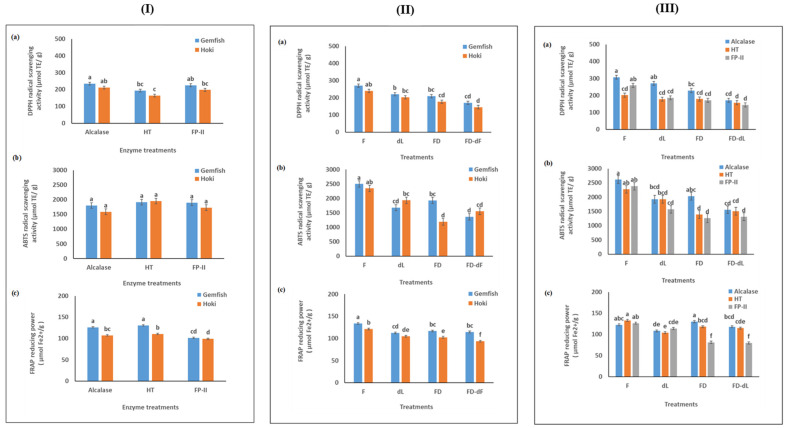
DPPH radical scavenging activity (**a**), ABTS radical scavenging activity (**b**), and Ferric (Fe^3+^) reducing power (**c**). Different enzyme treatments and fish roe homogenates (**I**), different enzyme treatments and fish roe homogenates (**II**), and different treatments and enzyme treatments (**III**). (a–f) indicating a significant difference in antioxidant activity among different treatments (*p* < 0.05). F = fresh, dL = delipidation, FD = freeze-dried, and FD-dL: freeze-dried with delipidation.

**Table 1 marinedrugs-22-00364-t001:** Fatty acid distribution (mg/100 g fresh weight and % of total fatty acids) of total lipids extracted from Hoki and Gemfish roes.

Sample	Hoki Roe (mg/100 g Fresh Weight)	Gemfish Roe (mg/100 g Fresh Weight)	Hoki Roe (% Fatty Acids)	Gemfish Roe (% Fatty Acids)
C14:0	210.7 ± 35.5 ^A^	62.1 ± 5.5 ^B^	3.0 ± 0.09 ^A^	1.4 ± 0.09 ^B^
C14:1	16.8 ± 1.8	ND	4.8 ± 0.2	ND
C15:0	28.4 ± 2.2 ^A^	9.7 ± 0.5 ^B^	0.4 ± 0.01 ^A^	0.2 ± 0.01 ^B^
C16:0	1020.8 ± 110.8 ^A^	568.3 ± 41.7 ^B^	14.8 ± 0.30 ^A^	12.5 ± 0.30 ^B^
C16:1	39.2 ± 6.3 ^A^	11.5 ± 0.8 ^B^	0.6 ± 0.06 ^A^	0.3 ± 0.01 ^B^
C17:0	55.0 ± 3.3 ^A^	23.6 ± 3.0 ^B^	0.8 ± 0.02 ^A^	0.5 ± 0.02 ^B^
C17:1	383.2 ± 94.1 ^A^	132.2 ± 6.5 ^B^	5.5 ± 0.29 ^A^	2.9 ± 0.29 ^B^
C18:0	163.2 ± 29.1 ^B^	199.3 ± 33.2 ^A^	2.4 ± 0.26 ^B^	4.4 ± 0.26 ^A^
C18:1 tn-9	63.3 ± 20.2 ^A^	21.0 ± 2.7 ^B^	0.9 ± 0.08 ^A^	0.5 ± 0.08 ^B^
C18:1 cn-9	1641.0 ± 292	1330.7 ± 61	23.7 ± 1.94 ^B^	29.4 ± 0.61 ^A^
C18:1 n-11	267.2 ± 43.3	192.8 ± 11.1	3.9 ± 0.25 ^B^	4.3 ± 0.28 ^A^
C18:2 tn-6	27.4 ± 6.1	ND	0.4 ± 0.05	ND
C18:2 cn-6	82.1 ± 8.2 ^A^	50.9 ± 5.2 ^B^	1.2 ± 0.03	1.1 ± 0.08
C18:3 n-3	31.1 ± 2.6	26.1 ± 2.8	0.5 ± 0.03 ^B^	0.6 ± 0.05 ^A^
C20:0	39.9 ± 6.4 ^A^	14.8 ± 3.3 ^B^	0.6 ± 0.12 ^A^	0.3 ± 0.08 ^B^
C20:1	324.5 ± 52.1 ^A^	165.1 ± 13.9 ^B^	4.7 ± 0.58	3.6 ± 0.21
C20:2	17.3 ± 2.3 ^A^	10.7 ± 1.4 ^B^	0.3 ± 0.02	0.2 ± 0.03
C20:3 n-3	63.0 ± 9.4 ^B^	80.2 ± 4.3 ^A^	0.9 ± 0.15 ^B^	1.7 ± 0.04 ^A^
C22:0	94.6 ± 16.5 ^A^	52.3 ± 5.6 ^B^	1.4 ± 0.30	1.2 ± 0.09
C22:1 n-9	23.5 ± 8.3	14.7 ± 0.9	0.3 ± 0.10	0.3 ± 0.02
C20:5 n-3 (EPA)	551.5 ± 36.6 ^A^	258.1 ± 18.1 ^B^	8.1 ± 0.90 ^A^	5.7 ± 0.41 ^B^
C24:0	18.7 ± 2.6 ^B^	17.1 ± 2.1 ^A^	0.3 ± 0.03 ^B^	0.4 ± 0.04 ^A^
C24:1	25.0 ± 5.4 ^A^	13.3 ± 0.7 ^B^	0.4 ± 0.05 ^A^	0.3 ± 0.004 ^B^
C22:5n-3 (DPA)	132.5 ± 16.2 ^B^	193.8 ± 14.5 ^A^	1.9 ± 0.30 ^B^	4.3 ± 0.20 ^A^
C22:6 n-3 (DHA)	1275.3 ± 67.0	972.1 ± 71.0	18.6 ± 1.63 ^B^	21.4 ± 0.59 ^A^
Unknown	233.8 ± 68.5 ^A^	111.5 ± 22.4 ^B^	3.4 ± 0.83	2.4 ± 0.40
SFA	1631.3 ± 0.58 ^A^	947.2 ± 0.21 ^B^	23.7 ± 1.35 ^A^	20.9 ± 0.62 ^B^
MUFA	2783.7 ± 2.83	1881.3 ± 0.64	40.2 ± 1.01	41.5 ± 4.40
PUFA	2180.2 ± 2.76	1591.9 ± 0.56	31.8 ± 1.31 ^B^	35.1 ± 1.12 ^A^
n-3	2048.4 ± 0.52	1535.3 ± 0.52	30.0 ± 0.91 ^B^	33.8 ± 0.91 ^A^
n-6	109.5 ± 0.05 ^A^	50.9 ± 0.05 ^B^	1.36 ± 0.03 ^B^	1.84 ± 0.03 ^A^
n-6/n-3	0.064	0.039	0.045	0.054

Each value represents the mean of triplicate samples. Different superscript letters (A, B) in the same row indicate a significant difference (*p* < 0.05). Results are presented as mean ± standard deviation. ND: no fatty acid was detected in the study. Abbreviation: EPA = eicosapentaenoic acid, DPA = docosapentaenoic acid, DHA = docosahexaenoic acid, SFA = Saturated fatty acids, MUFA = monounsaturated fatty acids, and PUFA = polyunsaturated fatty acids. FA = fatty acid.

**Table 2 marinedrugs-22-00364-t002:** Hoki and Gemfish roe lipids contain Phospholipid (% of total phospholipid, and µM/100 g fish roe fresh sample) contents.

Items	Hoki Roe (%)	Gemfish Roe (%)	Hoki Roe (µM/100 g Wet Tissue)	Gemfish Roe (µM/100 g Wet Tissue)
PA	6.1 ± 3.0 ^A^	7.3 ± 1.3 ^A^	59.1 ± 25.2	71.5 ± 25.2
PG	6.1 ± 1.1 ^A^	4.8 ± 0.3 ^B^	52.0 ± 20.3 ^A^	42.7 ± 6.8 ^A^
CL	2.6 ± 0.9 ^A^	3.4 ± 1.4 ^A^	20.7 ± 4.2 ^B^	28.4 ± 6.2 ^A^
LPE	ND	ND	ND	ND
LPS	27.4 ± 7.0 ^A^	33.3 ± 8.8 ^A^	247.0 ± 14.2 ^A^	323.7 ± 15.9 ^A^
SM	2.6 ± 0.3 ^A^	2.6 ± 0.1 ^A^	21.9 ± 5.9 ^A^	24.9 ± 5.6 ^A^
LDPG	10.1 ± 1.8 ^A^	9.2 ± 1.9 ^A^	86.0 ± 33.2 ^A^	83.2 ± 8.1 ^A^
PS	3.4 ± 1.4 ^A^	1.9 ± 1.1 ^A^	26.7 ± 7.0 ^A^	18.3 ± 7.2 ^A^
PI	4.1 ± 1.8 ^A^	3.6 ± 0.6 ^A^	34.1 ± 18.0 ^A^	33.5 ± 7.7 ^A^
PC	28.7 ± 6.2 ^A^	34.6 ± 6.5 ^A^	231.0 ± 42.6 ^B^	313 ± 28.2 ^A^

Different superscript letters (A, B) in the same row indicate a significant difference (*p* < 0.05). Results are presented as mean ± standard deviation. Abbreviations: PA = phosphatidic acid; LDPG = lyso-diphosphatidylglycerol; CL = cardiolipin; LPE = lyso-phosphatidylethanolamine; LPS = lyso-phosphatidylserine; SM = sphingomyelin; PG = phosphatidylglycerol; PS = phosphatidylserine; PI = phosphatidylinositol; PC = phosphatidylcholine; ND = not detected.

**Table 3 marinedrugs-22-00364-t003:** Total protein and specific activity (µmol/min/g substrate protein) of Alcalase, FP-II and HT protease preparations at pH (7.0) and temperature (45 °C). Results are presented as mean ± standard deviation.

Enzyme	Concentration (µL/mL or mg/mL)	U/mL Protease	Total Protein (mg/g)	Specific Activity µmol/min/g Substrate of Protein
*Hoki roe as a substrate*				
Alcalase	1	1077 ± 0.001	166.4 ± 61.7	6.5 × 10^6^ ± 6.0 × 10^3^
HT	1	710 ± 0.0219	669.9 ± 23.8	1.1 × 10^6^ ± 2.8 × 10^4^
FP-II	1	221 ± 0.01	427.4 ± 15.5	5.2 × 10^5^ ± 2.3 × 10^4^
*Gemfish roe as a substrate*				
Alcalase	1	698 ± 0.02	166.4 ± 61.7	4.2 × 10^6^ ± 9.0 × 10^4^
HT	1	633 ± 0.02	669.9 ± 23.8	9.5 × 10^5^ ± 2.7 × 10^4^
FP-II	1	314 ± 0.01	427.4 ± 15.5	7.4 × 10^5^ ± 2.1 × 10^4^

Alcalase commercially provided as a solution concentration of the prepared stock solution, the enzyme solution was prepared Alcalase (µL/mL), HT (mg/mL), and FP-II (mg/mL) as mentioned in second column. Alcalase (mg/g equivalent solution), HT (mg/g of dry powder), FP-II (mg/g of dry powder).

## Data Availability

All data generated or analyzed during this study are included in this published article.

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
