# Peer review of "Investigation of Antioxidant Activity of Protein Hydrolysates from New Zealand Commercial Low-Grade Fish Roes"

_marinedrugs, 2024, doi:10.3390/md22080364_

Round 1
Reviewer 1 Report
Comments and Suggestions for Authors
The authors gave a good example of studying the enzymatic hydrolysis of protein raw materials of marine origin on the example of fish roe, which allows obtaining hydrolysates with good antioxidant properties. The authors paid much attention to the kinetic patterns of obtaining hydrolysates with different enzyme concentrations. A large number of experiments are presented to compare the processes of enzymatic hydrolysis with three different enzyme preparations of raw materials subject to different pre-processing (delipidation, freeze-drying). Some drawback of the work is the lack of analysis of the obtained results using existing models of enzymatic hydrolysis of proteins.
There are the following minor remarks:
-The authors showed that increasing the enzyme concentration leads to an increase in the final degree of hydrolysis. This pattern is present in all experiments conducted by the authors. Apparently, this general pattern is characteristic of enzymatic hydrolysis of proteins. For example, it can be discussed from the point of view of existing models of proteolysis. For example, see the article: Int. J. Mol. Sci. 2022, 23(15), 8089.
- In many studies, calibration of the amino nitrogen determination method is carried out using leucine solutions. The authors used serine. What is the reason for choosing this amino acid? Or it doesn't matter which amino acid to take.
- The authors write that changes in protein morphology during pre-treatment may be the cause of differences in enzymatic hydrolysis. What protein characteristics are meant? What methods can be used to characterize the changes in morphology? Discuss this issue.
Author Response
There are the following minor remarks:
Comment 1-The authors showed that increasing the enzyme concentration leads to an increase in the final degree of hydrolysis. This pattern is present in all experiments conducted by the authors. Apparently, this general pattern is characteristic of enzymatic hydrolysis of proteins. For example, it can be discussed from the point of view of existing models of proteolysis. For example, see the article: Int. J. Mol. Sci. 2022, 23(15), 8089.
Response 1
Done. A statement to reflect this fact has been added in L299-301.
Comment 2
- In many studies, calibration of the amino nitrogen determination method is carried out using leucine solutions. The authors used serine. What is the reason for choosing this amino acid? Or it doesn't matter which amino acid to take.
Response 2
Serine was used following a published study [1] that investigated the hydrolysis of Hoki roe using the same three enzymes (Alcalase, HT, and FP-II) used in the present study. Using the same standard facilitates comparison and supports the discussion of the results.
Comment 3
- The authors write that changes in protein morphology during pre-treatment may be the cause of differences in enzymatic hydrolysis. What protein characteristics are meant? What methods can be used to characterize the changes in morphology? Discuss this issue.
Response 3
It is logical to suggest that exposure of protein to organic solvent and drying cause protein aggregation. These are well established effects. These were discussed in L291-292 and L361-366,
Reviewer 2 Report
Comments and Suggestions for Authors
In the manuscript (ID: marinedrugs-3146590), authors researched the antioxidant activity of protein hydrolysates from New Zealand commercial low-grade fish roes. In general, the content meets the requirements of Marine drugs. Therefore, I think the manuscript can be accepted and published in Marine drugs after a major revision. To ensure the publication readiness of the manuscript, the following issue needs to be addressed:
(1) The language needs to be improved both grammatically and scientifically. In addition, the writing of this manuscript is not standardized enough. It is recommended that the authors carefully revise the manuscript according to the requirements of the journal, including language issues and formatting issues.
(2) Title
--Line 3: Should be “protein hydrolysates” rather “roe protein hydrolysates”.
(3) Abstract
--Line 10: Should be “nutrient composition” rather “composition”.
--Line 14: Should be “proteases: Alcalase, HT, and FP-II.” rather “protease preparations: namely, Alcalase, HT, and FP-II.”.
--Line 14: Should be “commercial bacterial protease (HT), and fungal protease (FP-II)” rather “HT, and FP-II”.
--Line 18: Should be “Phosphatidylcholine was the main phospholipid” rather “Phosphatidylcholine was the main phospholipid content”.
--Line 19-20: Should be “its hydrolysate” rather “its hydrolysates”.
--Line 20: Please provide the full name of DPPHË™ and ABTS. When an abbreviation appears in the manuscript, write its full name first, and the abbreviation is written after the full name. Then, write abbreviations directly in other parts of the manuscript.
--Line 20-21: Should be “ferric reducing antioxidant power” rather “Ferric Reducing Antioxidant Power”.
--Line 23: Should be “p < 0.05” rather “p<0.05”. In addition, “p” should be in italics.
(4) Key words:
--Line 26: Should be “Antioxidant activity” rather “Antioxidant”.
--Line 26: Should be “Hoki roe; Gemfish roe” rather “fish roe; Hoki roe; Gemfish roe”.
--Suggest the authors to add this keyword “nutrient composition”.
(5) Introduction
--Line 30: Should be “Edible fish roes have” rather “Edible fish roe has”.
--Line 34-36: Enzymatic hydrolysis to generate bioactive peptides is a potential method to increase the value of low-grade fish roe since the intactness of the roe is not technical or quality issues. There are currently some studies on the use of fish roes to prepare peptides, and authors are advised to add their support for this content, such as ACE inhibitory peptides from Skipjack tuna roes, antioxidant peptides from protein hydrolysate of Skipjack tuna (Katsuwonus pelamis) roe.
--Line 37: “In vitro” should be in italics. In addition, there are similar errors in other parts of the manuscript, and authors are advised to check the whole manuscript carefully and correct these minor errors.
--37-39: In vitro enzymatic hydrolysis has been extensively used to generate bioactive peptides from high-protein foods and enhance their functional properties while preserving their nutritional values. Marine Medicine has also published some high-quality articles on enzymatic hydrolysis of marine biological proteins to prepare peptides, such as ACE inhibitory peptides from the collagens of monkfish, peptides from tuna cardiac arterial bulbs, antioxidant collagen peptides of Siberian sturgeon, etc., which are recommended for authors to apply here to support this content.
(6) Results and discussion
--Line 66, 67 and 167: Should be “Hoki and Gemfish roes” rather “Hoki and Gemfish roe”.
--Line 85: Should be “Hoki and Gemfish roes” rather “Hoki roe and Gemfish roe”.
--Line 97: Should be “DHA” rather “docosahexaenoic acid (DHA)”.
--Line 151-166: This content is not strongly related to this experiment, the authors only need to explain the enzyme activity in the materials and reagents.
--Line 197: Should be “degree of hydrolysis (DH)” rather “degree of hydrolysis”.
--Line 221: Should be “The roe protein hydrolysis of Hoki and Gemfish” rather “The protein hydrolysis of Hoki roe and Gemfish roe”.
--Line 268 and 309: The author needs to clarify the meaning of the letters "a-c" in the title of Figure 1-Figure 4 and Figure 7. In addition, it is suggested that the authors redraw Figure 1-Figure 4 and Figure 7 to increase the scale of horizontal and vertical coordinates.
--Line 461: FeSO4·7H2O. “4” and “2” should be subscripts
(7) Materials and Methods
--"3. Materials and Methods" has been omitted in this manuscript.
--Line 547: Should be “Alcalase” rather “Alcalase protease”.
--Line 555: Should be “2,2-dipheny1-1-picrylhydrazyl (DPPH)” rather “DPPH (2,2-dipheny1-1-picrylhydrazyl)”. In addition, there are similar errors in other parts of the manuscript, and authors are advised to check the whole manuscript carefully and correct these minor errors.
Comments on the Quality of English Language
Moderate editing of English language required.
Author Response
Comments and Suggestions for Authors
In the manuscript (ID: marinedrugs-3146590), authors researched the antioxidant activity of protein hydrolysates from New Zealand commercial low-grade fish roes. In general, the content meets the requirements of Marine drugs. Therefore, I think the manuscript can be accepted and published in Marine drugs after a major revision. To ensure the publication readiness of the manuscript, the following issue needs to be addressed:
Comment (1)
The language needs to be improved both grammatically and scientifically. In addition, the writing of this manuscript is not standardized enough. It is recommended that the authors carefully revise the manuscript according to the requirements of the journal, including language issues and formatting issues.
Response 1
Done. The English native-speaker co-author has been through the MS.
Comment (2)
Title--Line 3: Should be “protein hydrolysates” rather “roe protein hydrolysates”.
Response: Done
Comment (3)
Abstract --Line 10: Should be “nutrient composition” rather “composition”.
Response: Done
Comment (4) --Line 14: Should be “proteases: Alcalase, HT, and FP-II.” rather “protease preparations: namely, Alcalase, HT, and FP-II.”.
Response: Done
Comment (5) --Line 14: Should be “commercial bacterial protease (HT), and fungal protease (FP-II)” rather “HT, and FP-II”.
Response: Done
Comment (5) --Line 18: Should be “Phosphatidylcholine was the main phospholipid” rather “Phosphatidylcholine was the main phospholipid content”.
Response: Done
Comment (6) --Line 19-20: Should be “its hydrolysate” rather “its hydrolysates”.
Response: Done
Comment (7) --Line 20: Please provide the full name of DPPHË™ and ABTS. When an abbreviation appears in the manuscript, write its full name first, and the abbreviation is written after the full name. Then, write abbreviations directly in other parts of the manuscript.
Response: Done
Comment (8) --Line 20-21: Should be “ferric reducing antioxidant power” rather “Ferric Reducing Antioxidant Power”.
Response: Done
Comment (9) --Line 23: Should be “p < 0.05” rather “p<0.05”. In addition, “p” should be in italics.
Done. All the p in the manuscript have been changed.
Comment (10) (4) Keywords:
--Line 26: Should be “Antioxidant activity” rather “Antioxidant”.
--Line 26: Should be “Hoki roe; Gemfish roe” rather “fish roe; Hoki roe; Gemfish roe”.
--Suggest the authors to add this keyword “nutrient composition”.
Response: Done
Comment (11) (5) Introduction
--Line 30: Should be “Edible fish roes have” rather “Edible fish roe has”.
Response: Done
--Line 34-36: Enzymatic hydrolysis to generate bioactive peptides is a potential method to increase the value of low-grade fish roe since the intactness of the roe is not technical or quality issues. There are currently some studies on the use of fish roes to prepare peptides, and authors are advised to add their support for this content, such as ACE inhibitory peptides from Skipjack tuna roes, antioxidant peptides from protein hydrolysate of Skipjack tuna (Katsuwonus pelamis) roe.
Response: Done. The sentence has been changed to ‘‘Several studies have supported that protein hydrolysates derived from fish roe exhibit notable bioactivities, such as ACE inhibitory peptides discovered in Salmo salar and Carassius gibelio roes as well as antioxidant peptides found in the Skipjack tuna (Katsuwonus pelamis) roe’’ And references are added to line 38-40
--Line 37: “In vitro” should be in italics. In addition, there are similar errors in other parts of the manuscript, and authors are advised to check the whole manuscript carefully and correct these minor errors.
Response: Done
--37-39: In vitro enzymatic hydrolysis has been extensively used to generate bioactive peptides from high-protein foods and enhance their functional properties while preserving their nutritional values. Marine Medicine has also published some high-quality articles on enzymatic hydrolysis of marine biological proteins to prepare peptides, such as ACE inhibitory peptides from the collagens of monkfish, peptides from tuna cardiac arterial bulbs, antioxidant collagen peptides of Siberian sturgeon, etc., which are recommended for authors to apply here to support this content.
Response: Done. “Several studies reported that protein hydrolysates derived from fish roe exhibit notable bioactivities, such as ACE inhibitory peptides discovered in Salmo salar, Carassius gibelio, and collagens of Monkfish (Lophius litulon) Swim Bladders as well as antioxidant peptides found in the Skipjack tuna (Katsuwonus pelamis) roe [2–4].”
Comment (12) (6) Results and discussion
--Line 66, 67 and 167: Should be “Hoki and Gemfish roes” rather “Hoki and Gemfish roe”.
Done
--Line 85: Should be “Hoki and Gemfish roes” rather “Hoki roe and Gemfish roe”.
Done
--Line 97: Should be “DHA” rather “docosahexaenoic acid (DHA)”.
Done
--Line 151-166: This content is not strongly related to this experiment, the authors only need to explain the enzyme activity in the materials and reagents.
This section used different concentrations of the protease to optimize the concentration to be used in subsequent studies (section 2.5).
--Line 197: Should be “degree of hydrolysis (DH)” rather “degree of hydrolysis”.
Done
--Line 221: Should be “The roe protein hydrolysis of Hoki and Gemfish” rather “The protein hydrolysis of Hoki roe and Gemfish roe”.
Done
--Line 268 and 309: The author needs to clarify the meaning of the letters "a-c" in the title of Figure 1-Figure 4 and Figure 7. In addition, it is suggested that the authors redraw Figure 1-Figure 4 and Figure 7 to increase the scale of horizontal and vertical coordinates.
The following sentences were inserted into the legend of different figures separately
Figure 1-4: Letters a-c indicate significant differences among different samples prepared with varying enzyme concentrations at the same time point (p < 0.05).
Figure 7: (a-f) indicating a significant difference in antioxidant activity among different treatments (p < 0.05).
--Line 461: FeSO4·7H2O. “4” and “2” should be subscripts
Done
Comment (12) (7) Materials and Methods
--"3. Materials and Methods" has been omitted in this manuscript.
--Line 547: Should be “Alcalase” rather “Alcalase protease”.
Done
--Line 555: Should be “2,2-dipheny1-1-picrylhydrazyl (DPPH)” rather “DPPH (2,2-dipheny1-1-picrylhydrazyl)”. In addition, there are similar errors in other parts of the manuscript, and authors are advised to check the whole manuscript carefully and correct these minor errors.
Done
Round 2
Reviewer 2 Report
Comments and Suggestions for Authors
The authors have revised the manuscript (ID marinedrugs-3146590) and the quality of the manuscript has been improved accordingly. Therefore, I think that the manuscript can be accepted for publication in Marine drugs.